# Evaluation of a Push–Pull Strategy for Spotted-Wing Drosophila Management in Highbush Blueberry

**DOI:** 10.3390/insects15010047

**Published:** 2024-01-10

**Authors:** Cody C. Gale, Beth Ferguson, Cesar Rodriguez-Saona, Vonnie D. C. Shields, Aijun Zhang

**Affiliations:** 1Invasive Insect Biocontrol and Behavior Laboratory, Beltsville Agricultural Research Center, United States Department of Agriculture-Agricultural Research Service, Beltsville, MD 20705, USA; 2Phillip E. Marucci Center for Blueberry and Cranberry Research and Extension, Rutgers University, Chatsworth, NJ 08019, USA; beth.ferguson@rutgers.edu (B.F.); crodriguez@njaes.rutgers.edu (C.R.-S.); 3Department of Biological Sciences, Towson University, Towson, MD 21252, USA; vshields@towson.edu

**Keywords:** *Drosophila suzukii*, methyl benzoate, *Vaccinium corymbosum*, repellent, oviposition deterrent, behavioral control, natural product

## Abstract

**Simple Summary:**

The spotted-wing drosophila (SWD), *Drosophila suzukii*, is an invasive pest of soft-skinned fruits that has rapidly spread across the globe and causes hundreds of millions of dollars in crop losses worldwide. Today, the management of SWD heavily relies on the application of synthetic pesticides, which are potentially hazardous to humans, animals, other organisms, and the environment. A natural-product-based sustainable integrated pest management approach is urgently needed to reduce conventional synthetic pesticide usages. A promising behavior-based control method is the “push–pull” strategy, which uses a repellent to drive pests away from fruits (push) and towards SWD attractant-baited mass trapping devices (pull). Methyl benzoate, a naturally occurring chemical found in many plants and FDA-approved food additives, was found to be repellent to SWD in laboratory tests. In this study, we tested whether this compound could also be used to protect blueberries from SWD injury in the field. Our results demonstrated that methyl benzoate as a spatial repellent/oviposition deterrent can be deployed in blueberry fields to reduce the damage caused by SWD, although this repellent is not sufficient to act as a control strategy alone and will need to be integrated with other strategies to provide adequate protection to growers.

**Abstract:**

We evaluated a novel push–pull control strategy for protecting highbush blueberry, *Vaccinium corymbosum*, against spotted-wing drosophila (SWD), *Drosophila suzukii*. Methyl benzoate (MB) was used as the pushing agent and a previously tested SWD attractive blend of lure-scents was used as the pulling agent. MB dispensers (push) were hung in the canopy and lure-scent dispensers (pull) were hung in yellow jacket traps filled with soapy water around the blueberry bushes. Blueberries were sampled weekly, and any infestation was inspected by examining the breathing tubes of SWD eggs which protrude through the skin of infested fruit. The frequency of infestation, i.e., the proportion of berries infested with at least one egg, and the extent of infestation, i.e., the mean number of eggs in infested berries, were significantly reduced in treatments receiving MB dispensers as a pushing agent when infestation rates were very high. However, the mass trapping devices as a pulling agent did not provide comparable protection on their own and did not produce additive protection when used in combination with the MB dispensers in push–pull trials. We conclude that MB has the potential to be implemented as a spatial repellent/oviposition deterrent to reduce SWD damage in blueberry under field conditions and does not require the SWD attractant as a pulling agent to achieve crop protection.

## 1. Introduction

The spotted-wing drosophila (SWD), *Drosophila suzukii* (Matsumura) (Diptera: Drosophilidae), is a pest of soft-skinned fruits such as blackberries, blueberries, sweet cherries, table grapes, peaches, raspberries, and strawberries [1,2]. Native to East Asia, this species was introduced to the continental US in 2008, possibly in imported fruits [3,4,5]. It has also been introduced to Europe, South America, and Africa, and now causes millions of dollars in losses to fruit crops around the world [6,7,8,9,10,11,12]. 

Unlike most *Drosophila* species, commonly referred to as vinegar flies, which are attracted to rotting fruits, SWD infests fruits that are still ripening on the plant as the female SWD uses a heavily sclerotized serrated ovipositor to slice through the fruit skin and oviposit in the fruit flesh [13,14]. Eggs possess a pair of “breathing tube” filament structures which protrude through the incision yet are barely visible to the naked eye. Eggs typically hatch in around one day, and the fruit begins to collapse from the larvae feeding after only two days, making the window for growers to react to infestation very short [15]. 

The primary method of SWD control is calendar-based conventional pesticide applications [16,17,18]. Regular sprays have been an effective control method, but this is changing as SWD begins to develop resistance to these pesticides [19,20]. An integrated pest management (IPM) approach is more effective and sustainable than calendar-based pesticide applications, especially considering the negative environmental effects of those sprays [16]. One such approach, known as “push–pull”, uses a spatial repellent to drive pests away from fruits (push), combined with mass trapping or attract-and-kill techniques which use attractant lures to draw the pests into traps where they are killed (pull). 

Many studies have examined the effectiveness of using baited traps for monitoring adult SWD (e.g., [21,22,23,24,25,26,27,28,29,30]) but even some of the most effective baits are still less attractive than fresh fruits [31]. Traditional SWD traps frequently function by the uncontrolled release of volatiles, which produces a large initial burst of odors that quickly dissipates, and typically comprise apple cider vinegar, wine, yeast, and sugar in some combination, although there are some commercially available lures such as Scentry (Scentry Biologicals, Inc., Billings, MT, USA) and Trécé (Trécé, Inc., Adair, OK, USA) composed of acetic acid, ethanol, acetoin, and methionol [32,33].

Previous work by Larson et al. [33,34] assessed the selectivity for SWD of the above-mentioned commercial lures compared to traditional apple cider vinegar traps and a quinary blend of attractant components from fermented apple juice identified by Zhang and Feng [35]. This quinary blend of acetoin, ethyl octanoate, acetic acid, phenethyl alcohol, and ethyl acetate was found to be more selective for SWD than the other baits and showed promise as a monitoring tool and mass-trapping device in a 2-year field study in raspberry and blueberry in the US and several European countries [33]. 

In the tests aimed at identifying the attractive components of fermented apple juice, one compound, methyl benzoate (MB), demonstrated repellent/toxicant effects when SWD was exposed to MB by itself [36,37]. MB is a common volatile organic compound found in many floral and fruit aroma profiles; for example, it is the dominant scent component of feijoa fruits [38] and is one of the main floral volatiles emitted by snapdragon and petunia flowers to attract bumble bees for pollination [39,40,41]. Despite the toxic effects of MB on many pest species [42,43,44,45], it has been found to have low toxicity towards non-target beneficial insects such as honey bees (*Apis mellifera* L. [Hymenoptera: Apidae]), tomato bugs (*Nesidiocoris tenuis* Reuter [Hemiptera: Miridae]), and green lacewings (*Chrysoperla carnea* Stephens [Neuroptera: Chrysopidae]) [46,47,48]. 

The purpose of this study was to assess the potential for MB to function as a spatial repellent/oviposition deterrent to protect highbush blueberry (*Vaccinium corymbosum* L.) from SWD. We first conducted laboratory assays to determine if MB could reduce fruit infestation when emitted by a controlled-release dispenser. We then conducted push–pull field trials using MB dispensers as the pushing agent and the above-mentioned quinary blend of attractants as the pulling agent. We found significant reductions in the frequency and extent of infestation with this push–pull strategy that were driven by the repellent effects of MB, indicating its potential as an effective SWD control measure in blueberry fields. 

## 2. Materials and Methods

### 2.1. Chemicals

All chemicals were purchased from Sigma-Aldrich (St. Louis, MO, USA): methyl benzoate (MB), ≥99%, CAS 93-58-3; 3-hydroxy-2-butanone (acetoin, AT), ≥99%, CAS 512-86-0; ethyl octanoate (EO), ≥99%, CAS 106-32-1; acetic acid (AA), ≥99.7%, CAS 64-19-7; phenethyl alcohol (PE), ≥99%, CAS 60-12-8; and ethyl acetate (EA), ≥99.5%, CAS 141-78-6. 

### 2.2. Insects

SWD adults were provided from a colony maintained at the Philip E. Marucci Center for Blueberry and Cranberry Research, Rutgers University, Chatsworth, NJ, USA. The colony was maintained at the Beltsville Agricultural Research Center (Beltsville, MD, USA) on artificial diet in a laboratory growth chamber with the temperature held at 25 °C with a light cycle of 16:8 h L/D. The artificial diet was prepared following Dalton et al. [49]. Polystyrene Drosophila vials 28.5 × 95 mm (VWR, Radnor, PA, USA) were filled to ~1/3 their total volume with artificial diet before it solidified. Once the diet cooled and solidified, ~30–50 adults were added to each vial and the vial was stopped with a cellulose Drosophila vial plug (VWR, Radnor, PA, USA).

Adults used in laboratory assays were aged 3–8 days post emergence. This age range was achieved by a schedule that removed and killed all adults in a set of vials after breeding for 10 days. Three days after the removal of adults, all newly emerged adults were transferred to new vials. These adults, on the day of transfer aged 0–3 days old, were used in experiments a minimum of 3 days later and no more than 5 days later. Vials were discarded when the diet was spent after 4 weeks.

### 2.3. Oviposition Deterrence Laboratory Assays

To determine whether MB could act as a spatial, i.e., a non-contact, repellent/oviposition deterrent, we conducted 3 sets of caged no-choice assays with lab-reared SWD and store-bought blueberries in the presence of either a blank control dispenser (only containing polyester felt) or an MB dispenser loaded with 1000, 500, or 100 µL of MB as well as polyester felt.

MB dispensers were constructed following the procedures of Larson et al. [34] for making SWD attractive lure-scent dispensers. Specifically, a length of polyethylene tubing (6 MIL thickness, ULINE, Pleasant Prairie, WI, USA) was cut from the roll, one end was sealed using only the heat-sealing (no vacuum) function of a vacuum food sealer (FoodSaver, Atlanta, GA, USA), and a strip of 2.53 × 0.6 cm polyester felt (Grainger, Lake Forest, IL, USA) cut to ~5 cm was inserted for absorbing and retaining the chemicals. The dispenser was then weighed to obtain an empty weight. An amount of 1000, 500, or 100 µL of MB was pipetted onto the felt strip, the other end of the tubing was then heat-sealed, and the dispenser was weighed again to obtain a filled weight. In their experimentation, Larson and colleagues found that the length of tubing between the two seals of the dispenser influenced the emission rates of the chemicals, with a larger spacing producing higher emission rates (personal communication). For this reason, the spacing between the seals was decreased along with the loading volume. The space between seals was 5 cm for 1000 µL dispensers, 3 cm for 500 µL dispensers, and 2 cm for 100 µL dispensers, and the felt strips were reduced in size accordingly.

Four incubators (Vevor, Rancho Cucamonga, CA, USA) (internal dimensions 28 × 25 × 36 cm) were used simultaneously so that 2 replicates for each treatment (control and MB dispenser) were generated in every 24 h oviposition experiment. Data for controls and treatments were always generated simultaneously. The door window of each incubator was covered with aluminum foil, and a string of LED lights attached to an outlet timer switch, set to match the light cycle of the rearing incubator, was installed in each of them. The incubators were maintained at 26 °C and 75 mL of water was added to the humidity tray, resulting in a relative humidity of ~80%, as recorded by climate sensors (ThermoPro, Duluth, GA, USA) placed inside.

A collapsible mesh insect cage ~20 × 20 × 20 cm (RestCloud, Guangzhou, China) was set up in each incubator. Organic blueberries were purchased from the local supermarket, rinsed, and air-dried. Three blueberries were selected for each cage, inspected for damage, and were not used if they were damaged or lacked firmness. The 3 berries, calyx-up, were arranged triangularly within each cage such that they were equidistant from each other and the cage walls (Figure 1A). These sets of experiments took place over the course of multiple weeks and a fresh pack of blueberries was purchased at the beginning of each week. Typically, 2–4 experiments (4–8 independent replicates for control and treatment) were conducted each week.

SWD colony vials containing 3–8-day-old adults were chilled on ice to anesthetize flies. Flies were gently shaken onto Petri dishes which were also on ice, and then 15 male and 15 female flies were selected for each cage and placed in a temporary holding tube. After 5–10 min to allow the insects to recover, they were inspected for potential damage incurred during the transfer process and any injured flies were replaced.

Each dispenser, blank or MB, was held by a binder clip, with the wide part of the binder clip placed on the floor of the cage, such that a dispenser stood vertically in the middle of the berries in each cage without being in contact with a berry (Figure 1A). The flies were gently transferred from the temporary holding tubes to each cage and the cages were zipped closed.

After 24 h, cages were removed from the chambers and refrigerated for 5–10 min to anesthetize flies, and the blueberries were removed from the cages. Each berry was inspected with a dissection scope and the total number of eggs, as identified by the protruding breathing tube filaments (Figure 1B), was recorded.

MB dispensers were weighed before and after each 24 h experiment so that the total amount of MB emitted could be calculated. The 100 µL dispensers were empty by the end of an experiment, so a separate set of identically fabricated dispensers were weighed on an hourly basis to determine the release rates. Chambers were given at least two hours to air out after each experiment, after which no MB odor could be detected, before another experiment was started. Treatments assigned to each chamber were switched weekly.

### 2.4. Field Tests

The purpose of the field experiment was to test the effectiveness of a “push–pull” strategy to mitigate SWD damage to field-grown blueberries, with the “push” as MB emitted by dispensers and the “pull” as the quinary blend SWD attractant lure developed by Larson et al. [34] in insect traps containing drowning solution.

We conducted three field tests. The first took place between 22 June and 3 August 2022, at Butler’s Orchard, in Germantown, Maryland (MD), USA. Two tests were conducted in 2023. One was again conducted at Butler’s Orchard, from 12 June to 24 July 2023. The other was conducted at Wells’ Blueberry Farm, in Southampton, New Jersey (NJ), USA, from 29 June to 27 July 2023. Both locations were “U-Pick” farms which allow the public to harvest blueberries throughout the season. The results of the 2022 field test in MD led us to conduct the 2023 field tests with additional push and pull stimuli, explained in detail below.

The experimental design was of randomized complete blocks with 5 replicated blocks of 4 treatments: push (MB dispensers present, SWD attractant lures absent), pull (MB dispensers absent, SWD attractant lures present), push–pull (MB dispensers present, SWD attractant lures present), and control (MB dispensers absent, SWD attractant lures absent). The arrangement of plots was randomized within each block.

Blocks were set up with a ~35 m buffer spacing between blocks and row edges. Plots were ~5 m in length along a pair of adjacent rows, with a ~1 m spacing between bushes, such that each plot contained 10 bushes to be sampled, with ~10 m buffers between plots (Figure 2). SWD traps were hung atop bamboo stakes ~2 m in height which were staked into the ground 2 rows away from the sampling rows. In other words, on either side of the sampling rows was a gap row containing neither MB dispensers nor traps, followed by a row with traps spaced equally along the 5 m plot length (in 2022, the traps were hung at the corners of the plots and, in 2023, the additional 2 traps were hung in the middle). Row spacing was ~2.5 m, so traps were hung ~5 m from sampling bushes. A diagram block design is provided in Figure 2.

As the pushing agent, MB dispensers were constructed following the same procedures of the laboratory bioassay. The only difference is that the thinner polyethylene tubing (2 MIL thickness, ULINE) was used to increase the MB release rate in the field. The polyethylene tubing was cut to a length of 12 cm, with the space between seals being 10 cm. Dispensers were filled with 8 mL of MB. For the first field test, which occurred in 2022, a single MB dispenser was prepared for each bush of a push or push–pull plot. For the tests that occurred in 2023, three MB dispensers were prepared for each bush of a push or push–pull plot, for a total of 24 mL MB per bush, and the three dispensers were hung together in the center of the upper canopy by situating them in the clip loop of a plastic sign holder stake (A.M. Leonard, Piqua, OH, USA, product discontinued) (Figure 1C). Blank dispensers were prepared for pull and control plots.

MB dispensers, prepared as described above, were assessed for the release rate of MB and the duration of time until they became empty under field conditions. Release rate tests were conducted in June 2022, with 24 h mean temperatures that ranged from 19.4 to 34.6 °C, mean relative humidities that ranged between 29.8 and 79.8%, and mean wind speeds that ranged between 1.6 and 11.5 mph. Release rate was calculated from daily weights of three replicated dispensers. MB was released at a mean rate of 19.6 mg/h. With 8 mL loaded (8.7 g), the dispensers lasted 18 days on average (linear regression of weight remaining in g(y) and time in days (x): y = −0.47x + 8.7, R^2^ = 0.99, N = 3). To account for the unpredictability of field conditions, dispensers were replaced every 14 days for the experiment.

As the pulling agent in all field tests, SWD attractant lure dispensers were prepared as described by Larson et al. [34] (Table 1). Tubing was cut to a length of 10 cm, with the space between seals being 8 cm, such that the remaining 2 cm could be used for hanging.

The quinary blend SWD attractant lure was hung in commercially available Victor^®^ yellow jacket and flying insect traps (Woodstream Corporation, Lancaster, PA, USA) filled with ~300 mL of tap water containing surfactant (4 mL/gallon “free and clear” natural dish liquid, Seventh Generation, Inc., Burlington, VT, USA) as a drowning solution (Figure 1D). The lures were hung inside the traps on hooks fashioned from plastic-coated paperclips. In the 2022 field test, 4 SWD traps were hung per plot. In the 2023 field tests, 6 SWD traps were hung per plot. Dispensers were replaced every 14 days [34]. Blank dispensers containing only polyester felt were prepared for traps in push and control plots.

All traps were emptied into sample cups and the drowning solution was replaced every 7 days (Table 2). The trap catch was drained, transferred to a Petri dish, and viewed with a dissection scope to obtain a count for SWD caught. SWD adults were identified following van Timmerman et al. [50]. The sampling date, block number, plot treatment, number of male and female SWD, and total number of other fruit flies were recorded for each trap.

Berries were sampled every 7 days, on the same day as traps, starting 7 days after materials were set up in the field (Table 2). Two berries were sampled per bush for a total of 20 berries per plot. Each of the 2 berries was sampled from 2 different branches, one from an inner canopy branch and one from an outer canopy branch. This was performed to test for differences in SWD oviposition based on proximity to the inner canopy where the MB dispensers were hung. Berries were only sampled if they represented berries that would be suitable both as an SWD oviposition substrate and as berries that would be purchased by consumers. They were ripe, firm, and undamaged to the naked eye. SWD will not oviposit in green, very unripe berries. Berries that lacked firmness could have already started to be consumed by SWD larvae. Noticeable damage to a berry may have influenced SWD preference. By selecting only ripe, firm, and undamaged berries, we avoided bias towards or against SWD damage and obtained samples representative of what U-pick customers would harvest. Berries were inspected for SWD eggs within 24 h using a dissection scope. The date of sampling, block number, plot treatment, position (inner or outer canopy), and number of eggs were recorded for each berry.

## 3. Statistical Analysis

### 3.1. General

All statistics were performed in R version 4.2.2 [51]. Statistical significance was considered at α = 0.05, though marginal significance was considered at α = 0.1, and a result was considered highly significant at α = 0.01.

### 3.2. Oviposition Deterrence Laboratory Assays

Each cage was treated as a replicate with the mean number of eggs per berry as the independent sample. Independent replicates (equivalent N for control and treatment) for each experiment are as follows: 1000 µL N = 20, 500 µL N = 22, and 100 µL N = 24. Data were analyzed by a generalized linear mixed-effects model of the negative-binomial distribution using the glmer.nb function from the package lme4 [52]. The mean number of eggs (rounded to the nearest integer) was treated as the response to treatment (control or MB) and dispenser volume (1000, 500, 100 µL) as fixed effects, while controlling for the date and incubator as random effects. Model diagnostics were performed with the package DHARMa [53]. The significance of the fixed effects was determined with a type II Wald Chi-square test using the Anova function from the package car [54]. Pairwise comparisons were obtained with the emmeans function from the package emmeans [55].

### 3.3. Field Tests

Each block was treated as a replicate and the plot was treated as an independent sample. We planned to have 5 blocks sampled each week for all treatments, but the number of suitable berries for sampling decreased as the season progressed due to continual harvesting by U-pick customers, and the number of independent replicates for each treatment are as follows: 2022 MD control N = 25, push N = 23, pull N = 25, push–pull N = 26; 2023 MD control N = 24, push N = 24, pull N = 23, push–pull N = 22; 2023 NJ control N = 17, push N = 19, pull N = 17, push–pull N = 16.

Data were analyzed in terms of the frequency of infestation, i.e., the proportion of berries infested with at least one egg, and the extent of infestation, i.e., the mean number of eggs in infested berries. Both of these terms are important since the presence of a single larva renders a fruit unmarketable. These measures were treated as the responses to treatment (control, push, pull, push–pull), position (inner or outer canopy), and the interaction term of these factors. The frequency and extent of infestation were analyzed separately.

To assess the frequency of infestation, berries with at least 1 egg were labeled “successes” and berries with zero eggs labeled “failures.” This ratio of successes to failures was used as the response to treatment, position, and their interaction as the fixed effects while controlling for sampling date and block number as the random effects in a mixed-effects binomial logistic regression using the glmer function from the package lme4 [52].

To assess the extent of infestation, the mean number of eggs per infested berry (rounded to the nearest integer) was treated as the response to treatment, position, and their interaction as the fixed effects while controlling for sampling date and block number as the random effects in a generalized linear mixed-effects model of the negative-binomial distribution using the glmer.nb function from the package lme4 [52].

The trap catch was analyzed with a generalized linear mixed-effects model of the negative binomial distribution with count as the dependent variable, fly (male SWD, female SWD, or other fruit fly) and treatment as the independent variables, and date and block as random effects.

Model diagnostics were performed with the package DHARMa [53], the significance of the fixed effects was determined with a type II Wald Chi-square test using the Anova function from the package car [54], and pair-wise comparisons were obtained with the emmeans function from the package emmeans [55].

## 4. Results

### 4.1. Oviposition Deterrence Laboratory Tests

The 1000 µL dispensers emitted an average of ~15 mg MB/h (linear regression of weight remaining in g(y) and time in days (x): y = −0.36787x + 1.03761, R^2^ = 0.974, N = 3). The 500 µL dispensers emitted an average of ~11 mg MB/h (linear regression of weight remaining in g(y) and time in days (x): y = −0.27167x + 0.52167, R^2^ = 0.9684, N = 3). The 100 µL dispensers emitted an average of ~7 mg MB/h (linear regression of weight remaining in g(y) and time in h (x): y = −0.0066552 + 0.1104622, R^2^ = 0.99, N = 3). The treatment significantly influenced oviposition levels (χ^2^ = 31.093, Df = 1, *p* < 0.0001), and so did dispenser volume, i.e., release rate (χ^2^ = 12.350, Df = 2, *p* = 0.0021). In each set of tests, berries in the presence of an MB dispenser experienced significantly less oviposition than those in control cages (*p* < 0.0001). Berries in control cages were observed to have an average of 10 eggs in the 1000 µL tests, 14 eggs in the 500 µL tests, and 15 eggs in the 100 µL tests. Berries in MB cages were observed to have an average of 1 egg in the 1000 µL tests, 4 eggs in the 500 µL tests, and 10 eggs in the 100 µL tests (Appendix A).

### 4.2. Field Tests

In the 2022 MD field test, the overall frequency of infestation was 32% (Appendix A) and the overall mean extent of infestation was 4.17 eggs per infested berry (Appendix A). The frequency of infestation for control plots was 29.3%, while push plots had 24.1%, pull had 31.5%, and push–pull had 30.5% (Figure 3A). Although the frequency of infestation was not significantly affected by treatment (Table 3), the extent of infestation was affected by treatment with marginal significance (Figure 3B, Table 3). Infested berries in control plots had a mean of 3.47 eggs, while that for push was 3.15, that for pull was 4.69, and that for push–pull was 3.2 (Figure 3B). The effect of position was highly significant on both the frequency and extent of infestation, while the interaction of treatment and position did not significantly affect either (Table 3). Inner-canopy berries had a frequency of infestation of 32.7%, while outer-canopy berries saw 25.1% (Appendix A). The extent of infestation for inner-canopy berries was a mean of 4.35 eggs, and for outer-canopy berries, it was 2.94 eggs (Appendix A).

Given the marginally significant effect of treatment on the extent of infestation in the 2022 MD field test, we conducted the 2023 field tests using 300% more MB dispensers and 50% more traps to determine if a greater amount of push and pull stimuli could further reduce SWD infestation.

In the 2023 MD field test, the overall frequency of infestation was 14% (Appendix A) and the overall mean extent of infestation was 1.46 eggs per infested berry (Appendix A). The frequency of infestation and extent of infestation were not significantly affected by treatment (Table 3). Control plots had a frequency of infestation of 7.8%, while that for push was 10.3%, that for pull was 11.0%, and that for push–pull was 9.0% (Figure 3A). The extent of infestation for all treatments in the 2023 MD field test was ~1 egg per infested berry (Figure 3B). Position again had a highly significant effect on both the frequency and extent of infestation, while the interaction of treatment and position did not, with inner-canopy berries seeing a frequency of 13.1% while outer-canopy berries saw 6.7% (Table 3, Appendix A).

In the 2023 NJ field test, we observed the overall rate of infestation to be 71% (Appendix A) and the overall extent of infestation to be 4.17 eggs per infested berry (Appendix A). Control plots had a frequency of infestation of 83.3%, while that for push was 56.5%, that for pull was 77.8%, and that for push–pull was 65.0% (Figure 3A). In contrast to the other tests, the treatment had a highly significant effect on both the frequency and extent of infestation (Table 3). Infested berries in control plots had a mean of 5.38 eggs, while that for push was 2.96, that for pull was 3.59, and that for push–pull was 3.00 (Figure 3B). Further contrasting the other tests, position had only a marginally significant effect on the frequency of infestation, but no significant effect on the extent (Table 3). Inner-canopy berries had a frequency of infestation of 69.7% and an extent of 3.55 eggs, while outer-canopy berries saw a 73.8% frequency and an extent of 3.69 eggs (Appendix A). The interaction of treatment and position again did not have a significant effect on either (Table 3).

Unbaited traps in control and push plots did not catch SWD in any of the field tests; thus, they were not included in the analysis. Lure-baited traps from the pull and push–pull plots caught an overall average of 9 (±2) female SWD, 12 (±2) male SWD, and 41 (±8) other fruit flies per trap. The treatment did not have a significant influence on the number of flies caught (χ^2^ = 1.1596, N = 140, Df = 1, *p* = 0.2816), but the numbers of male SWD, female SWD, and other fruit flies were significantly different from one another (χ^2^ = 285.363, Df = 2, *p* < 0.0001). Details of weekly trap catches and overall berry infestation are provided in Figure 4.

## 5. Discussion

The purpose of this study was to evaluate MB, an emerging botanical pesticide, as a potential spatial repellent/oviposition deterrent to reduce SWD infesting blueberries. This study consisted of three stages: beginning with preliminary laboratory assays to determine if MB could reduce infestation by SWD on store-bought blueberries, followed by a field test at one location in 2022, and then further field testing at two locations in 2023. Laboratory assays showed clearly that MB dispensers which emitted at least ~7 mg/h were sufficient to significantly reduce SWD oviposition on store-bought blueberries in a controlled environment (Appendix A). These 7 mg/h dispensers represent the minimum level of MB release that was tested in the lab, and this release rate is more than eight-fold less than the emissions of the dispensers used in field tests that followed.

We deployed MB dispensers in the field as the pushing agent in a push–pull control strategy, with the pulling agent being lure-baited mass trapping devices previously shown to be relatively selective for trapping SWD. From the three field tests conducted, we found that MB dispensers can be used as a repellent/oviposition deterrent to SWD in blueberry fields, significantly reducing the frequency of infestation, i.e., the proportion of fruits infested with at least one egg, and the extent of infestation, i.e., the number of eggs in infested fruits compared to controls, but only when infestation rates are very high.

In the first field test conducted in 2022 at the MD site, we deployed a single ~20 mg/h MB dispenser per blueberry bush and found that the frequency of infestation in push plots was about 5% less than in controls plots. Push and push–pull treatments experienced lesser extents of infestation than pull plots, but not controls, with marginal significance (Figure 3, Table 3, Appendix A). For these reasons, the testing in 2023 used additional push and pull stimuli under the hypothesis that greater concentrations of these chemicals in the field could further our understanding of the behavior-modifying components of this control strategy.

The 2023 field tests, one of which took place in MD and the other in NJ, experienced very different infestation rates. At the MD site in 2023, the overall frequency of infestation was less than half of what it was in 2022, at 14% compared to 32%, while the NJ site saw a 71% infestation (Appendix A). The 2023 MD test did not show any significant differences between treatments in the frequency or extent of infestation, while the 2023 NJ test saw highly significant differences among treatments for both measures (Figure 3, Table 3, Appendix A). Our data produced an unexpected trend between the level of pest damage and the push–pull effects, with overall infestation rates of 14% indicating no significant effect, 32% revealing a marginally significant effect, and 71% exhibiting a highly significant effect (Figure 3, Table 3). Since only the one field test produced significant results for treatment differences, further testing in high-infestation scenarios is needed to better understand this trend, ideally with frequencies of infestation between 32 and 71%.

Why the push–pull strategy used in our study showed the greatest effect under the highest infestation levels needs further investigation to be understood. An inverse relationship between the effectiveness of an IPM strategy that employs olfactory stimuli and pest density can be found in mating disruption tactics. Mating disruption uses sex pheromones to mislead and confuse individuals seeking mates, leading to fewer successful pairings and reducing the number of fertilized eggs produced in that population. In competitive mating disruption, the effectiveness of this control method is highly pest-density-dependent and provides the greatest control when pest densities are low [56,57]. Mating disruption techniques differ from the push–pull technique in that the olfactory stimuli causing the behavior modification are highly species-specific in mating disruption, whereas the repellent used in push–pull may not be. MB, for example, has been previously reported as a repellent to the brown marmorated stink bug *Halyomorpha halys* Stål (Hemiptera: Pentatomidae), the white fly *Bemisia tabaci* Gennadius (Hemiptera: Aleyrodidae), and the spider mite *Tetranychus urticae* Koch (Acari: Tetranychidae) [36,58,59].

This study also examined the effectiveness of a pulling agent, a quinary blend of odorants tailored to SWD attraction, in controlling this pest. The number of SWD caught in lure-baited traps did not correlate with the levels of berry infestation observed (Figure 4). In the two MD field tests, there was a general trend for both the trap catch and berry infestation to be low at the beginning of the season and increase as the season progressed, but the greatest numbers of trap catches (Figure 4A) did not correspond to the highest levels of infestation (Figure 4B). The greatest number of female SWD trapped during a field test were caught in the 23 MD test (Figure 4A), which had the lowest levels of infestation overall (Figure 3A,B, Appendix A). In the 23 NJ test, the total number of infested berries decreased during the course of the season because the number of berries available for sampling decreased due to U-pick customer harvest, but the mean number of eggs per infested berry can be seen to decrease each week (Figure 4A) even though trap catches trended upward (Figure 4B). A general lack of correlation between trap catches and fruit infestation is a notable trend in SWD research [16].

The trends in our data for the pulling component of this push–pull strategy indicate that these mass trapping devices can reduce the extent of infestation compared to controls when infestation rates are very high; however, our pull component tended to lead to more infestation when used alone compared to when used in conjunction with the push, or when compared to a push agent used alone (Figure 3B). In other words, the protection provided by this pulling agent is not additive to the protection offered by the pushing agent and may not offer protection against SWD when used alone. This trend has been observed in similar studies as well. For example, Wallingford et al. [60] tested a push–pull strategy to control SWD in raspberry using 1-octen-3-ol as the pushing agent and the commercially available Scentry Lure-baited traps as the pulling agent. They found that the pull treatments alone reduced oviposition in laboratory assays but led to increased levels of infestation in the field when used alone. Moreover, 1-octen-3-ol when used as the pushing agent was more effective at reducing infestation when used alone than when used in conjunction with the pulling agent, even though the push–pull combination appeared to offer additive protection in the laboratory assays [60].

Another similar study was conducted by Cha et al. [61], in which they used 2-pentylfuran as a pushing agent against SWD in raspberry. In preliminary trials, they found that 2-pentylfuran reduced SWD attraction to lures by 91.1% compared to 1-octen-3-ol. In field trials, they used 2-pentylfuran dispensers with emission rates of 14 mg/h and found that these dispensers reduce SWD infestation in raspberry clusters by 56% compared to untreated plots. A pulling agent was not used in that study, lending further evidence to the idea that, in the case of SWD, the pull is not a necessary component. Thus, attractant devices are useful for monitoring SWD populations but are not necessary for achieving reduced fruit damage.

We note that the study by Cha et al. [61], like the study by Wallingford et al. [60], used the emergence of SWD from sampled fruits as their measure of infestation levels. While this is a common and practical approach, we suggest that the methods we used in the present study provide additional information about the effectiveness of an experimental control method. We hypothesize that SWD control methods that use spatial repellents/oviposition deterrents may alter the reproductive environment for flies in such a way that they are pushed from berries where the pushing agent is strong towards berries where the pushing agent is weak, creating a scenario where those less protected berries by a repellent act as a sink for SWD infestation. A scenario such as this could be useful for growers, as they could avoid harvesting the berries that act as sinks because the heavily infested berries will quickly lose their firmness. Thus, having data on both the frequency and extent of infestation can provide more information on exactly how these repellents are altering the chemical environment of stimuli that control oviposition behavior by SWD.

We also examined the effects of this push–pull strategy on the positional preferences of SWD oviposition under the hypothesis that the MB dispensers might repel. SWD is known to have a higher population of adults and fruit infestations in the inner canopy [16,62]. In our experiments, inner canopy berries had significantly greater frequencies and extents of infestation than outer canopy berries in both field tests in MD but not in the NJ field test (Appendix A). We hypothesize that this difference is due to the high levels of infestation that were observed in the NJ study compared to the other two and that, when the infestation is sufficiently high, resource competition might lead to this loss in an observable preference. There was no interaction between treatment and position, indicating that the protection offered by the presence of MB dispensers was not limited to the inner canopy berries nearest the dispensers but extended to the outer canopy berries as well.

MB is an FDA- and European Union-approved food additive [63,64] and is safe for human consumption. It has previously been demonstrated to be an effective ovicide and larvicide for SWD-infested blueberries [36]. In that study, SWD-infested blueberries were dipped in 1% aqueous MB solution, resulting in 100% SWD mortality, thus demonstrating the curative activity of MB. Curative activity is the lethal action of an insecticide against the target pest post infestation [65]. Of the commonly used conventional insecticides, neonicotinoids, organophosphates, and spinosyns have demonstrated curative activities against SWD in blueberries [66,67]. Neonicotinoids, however, are not recommended for SWD management [17], and SWD resistance development to spinosyns and malathion, an organophosphate, has been documented [19,20,68]. The curative action of MB against SWD in blueberry [36] combined with the repellent effect observed in our study indicate that this botanical insecticide has the potential as a promising alternative to be used for fruit protection pre- and post infestation.

Our current study adds to a growing body of evidence demonstrating MB as a natural product that can not only kill but repel pests such as SWD, with great potential to be utilized in the field to reduce crop infestation. The olfactory responses of flies change over the course of the season as fruit availability changes, and in response to their physiological and reproductive status [69,70,71]. Subsequent research should seek to determine if the choice of repellent/oviposition deterrent determines its performance at different stages in the field season, and if a rotation or combination of these repellents might perform better than one individually, or if a variation can prevent SWD from losing sensitivity to a single compound. The mechanism of action for MB as a pesticide and exactly how MB functions as a repellent have yet to be explored. Future research should also examine the mechanisms underlying the repellent/oviposition deterrent effects observed in our study.

## Figures and Tables

**Figure 1 insects-15-00047-f001:**
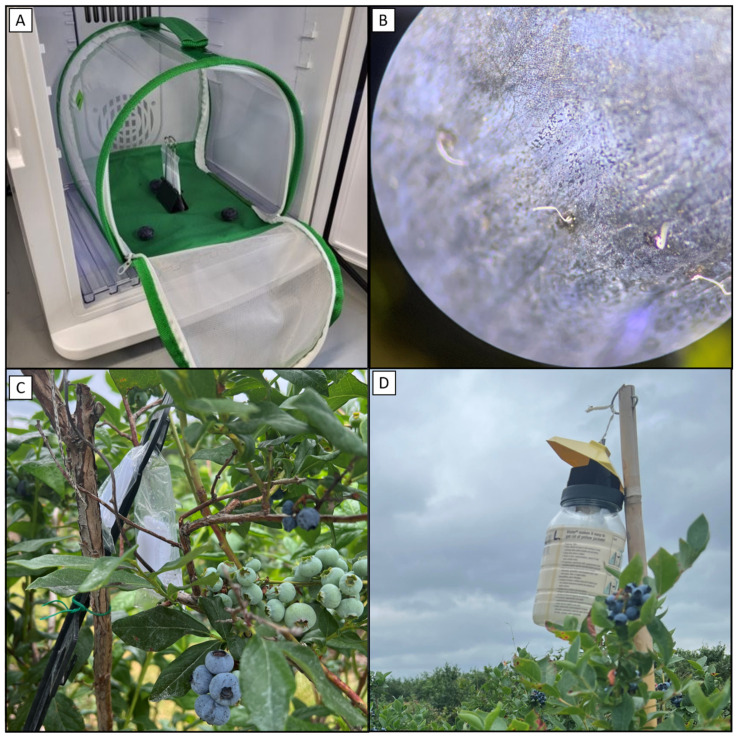
(**A**) Example of a test cage set up for the oviposition deterrence laboratory assays. The blank or MB dispenser is held by a binder clip which acts as a stand in the middle of the cage. Three blueberries are placed triangularly around the dispenser. (**B**) Photo taken through the eyepiece of a dissection scope showing the breathing tube filaments of SWD eggs which protrude through the fruit skin. The breathing tubes of 4 eggs can be seen. Two filaments extend from each egg but are often stuck together appearing as one. (**C**) Three MB dispensers, which are sealed together at their tops, are hanging in the canopy of a blueberry bush. They are secured to a black plastic sign stake which is twist-tied to a branch. Clear liquid MB can be seen in the dispensers. (**D**) A Victor^®^ (Woodstream Corporation, Lancaster, PA, USA) yellow jacket trap containing SWD attractant lure dispensers hangs above a blueberry bush from atop a bamboo stake.

**Figure 2 insects-15-00047-f002:**
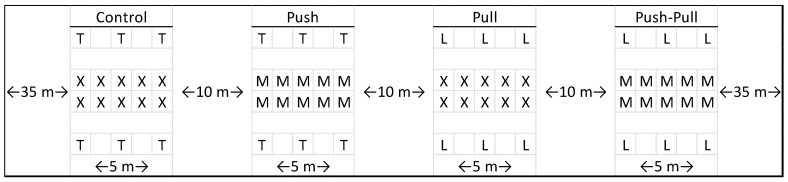
Not to scale. Diagram of a complete block containing the 4 treatment plots. X: blueberry bush with blank dispenser; T: trap with blank dispenser; M: blueberry bush with MB dispenser; L: trap with quinary blend lure dispensers. This diagram represents the tests that occurred in 2023. The test of 2022 only included 4 traps at the corners, and not the additional 2 in the centers.

**Figure 3 insects-15-00047-f003:**
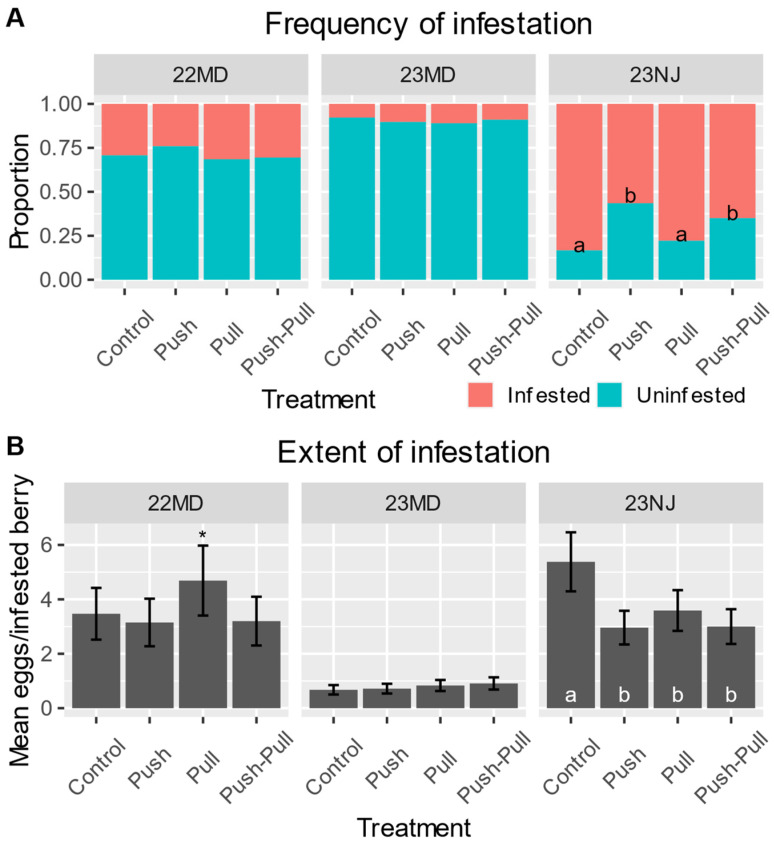
In panels with lower-case lettering, treatments assigned different letters are highly significantly different at α = 0.01. An asterisk indicates marginally significant differences between treatments at α = 0.1. Statistics are presented in Table 3 and Appendix A. (**A**) The frequency of blueberries being infested with at least one SWD egg for each treatment in the field. Proportions are estimated by the logistic regression models. (**B**) The mean (±SE) number of SWD eggs per infested blueberry for each treatment in the field. Means are estimated by the generalized linear models.

**Figure 4 insects-15-00047-f004:**
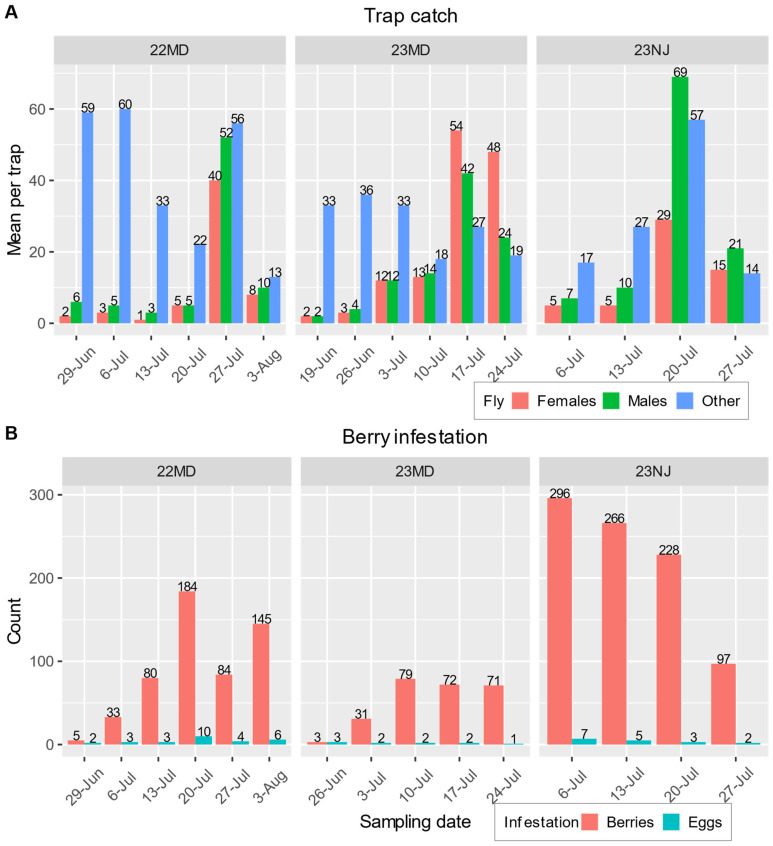
(**A**) Mean number of female SWD (orange), male SWD (green), and other fruit flies (blue) caught in lure-baited traps on the day of sampling for every week in each field test. (**B**) The total number of infested berries (orange) and the mean number of eggs in infested berries (blue) on the day of sampling for every week in each field test (no infestation was detected on 19 June 2023 in MD).

**Table 1 insects-15-00047-t001:** Materials used to construct the dispensers used in the field tests.

Dispenser	Compound	Loading Volume (Blend Proportion)	Polyethylene Tubing Thickness
Push (repellent)	Methyl benzoate	8 mL	2 MIL
Pull (attractant)	Acetic acid	5 mL	2 MIL
	Ethyl acetate	5 mL	6 MIL
	Phenethyl alcohol	1 mL	2 MIL
	Ethyl octanoate, Acetoin	2 mL (1:1)	2 MIL

**Table 2 insects-15-00047-t002:** Outline of field activities performed during each field test. MD = Maryland; NJ = New Jersey.

Field Test	Date	Activities
22 MD	22 June 2022	Set up traps and dispensers
	29 June 2022	Sampled traps and berries
	6 July 2022	Sampled traps and berries, replaced dispensers
	13 July 2022	Sampled traps and berries
	20 July 2022	Sampled traps and berries, replaced dispensers
	27 July 2022	Sampled traps and berries
	3 August 2022	Sampled traps and berries
23 MD	12 June 2023	Set up traps and dispensers
	19 June 2023	Sampled traps and berries
	26 June 2023	Sampled traps and berries, replaced dispensers
	3 July 2023	Sampled traps and berries
	10 July 2023	Sampled traps and berries, replaced dispensers
	17 July 2023	Sampled traps and berries
	24 July 2023	Sampled traps and berries
23 NJ	29 June 2023	Set up traps and dispensers
	6 July 2023	Sampled traps and berries
	13 July 2023	Sampled traps and berries, replaced dispensers
	20 July 2023	Sampled traps and berries
	27 July 2023	Sampled traps and berries

**Table 3 insects-15-00047-t003:** Analysis of deviance table for the models of the frequency and extent of infestation for all three field tests combined. Significance codes: single asterisk (*), marginally significant at α = 0.1; triple asterisk (***), highly significant at α = 0.01.

Field Test	Response	Factor	χ^2^	Df	*p*
2022 MD	Frequency of infestation	Treatment	3.958	3	0.266032
		Position	7.5626	1	0.005959 ***
		Treatment:Position	0.5013	3	0.918615
	Extent of infestation	Treatment	6.3418	3	0.09612 *
		Position	21.9587	1	<0.0001 ***
		Treatment:Position	1.0299	3	0.79401
2023 MD	Frequency of infestation	Treatment	4.9245	3	0.1774
		Position	26.8748	1	<0.0001 ***
		Treatment:Position	1.7799	3	0.6193
	Extent of infestation	Treatment	3.3753	3	0.3372961
		Position	12.6701	1	0.0003715 ***
		Treatment:Position	4.4535	3	0.2164736
2023 NJ	Frequency of infestation	Treatment	61.3336	3	<0.0001 ***
		Position	3.0326	1	0.08161 *
		Treatment:Position	3.4441	3	0.32808
	Extent of infestation	Treatment	38.7119	3	<0.0001 ***
		Position	0.3125	1	0.5762
		Treatment:Position	0.5517	3	0.9074

## Data Availability

The data presented in this study and the R code used for statistical analysis are available on request from the corresponding author.

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
