# Peer review of "Evaluation of a Push–Pull Strategy for Spotted-Wing Drosophila Management in Highbush Blueberry"

_insects, 2024, doi:10.3390/insects15010047_

Round 1

Reviewer 1 Report

Comments and Suggestions for Authors

The manuscript by Gale et al. entitled “Evaluation of a push-pull strategy for spotted-wing Drosophila management in highbush blueberry orchards” tests MB (methyl benzoate), a volatile flower compound, as a repellent in “push” and commercial traps with attractant as the “pull”. The authors test treatments of control (no chemicals), push (MB), pull (not clear which attractant was used in each exp. and dosage – line 239), and push-pull (combination). They test MB in the laboratory and show it has repellent and/or toxic effects in a confined cage inside an environmental chamber because the number of eggs is lower in the blueberry fruits. This is not very important, but the field results are a serious attempt at testing the push, pull, and push-pull management methods. The statistics appear to be advanced. Unfortunately, the work in 2022 was in moderately-infested plots (~30%) and there were no significant (statistically) differences among the treatments (Table 1, Fig. 3). Also, the figure 3a indicates that there are hardly any differences frequency of infestation in 2022 (Maryland) between control and the MB treatments (push and push-pull) as well as pull (commercial trap). This was also true of 2023 in Maryland which had a relatively low frequency of infestation and extent of infestation (both 7 to 13%). However, in New Jersey in 2023 the infestation frequency (~83%) and extent of infestation (~70%) were the highest of the three study periods. In this NJ23 there were some significant differences of the push, push-pull, and pull treatments that were roughly 50 to 60% of the control. This might indicate that the three treatments might be effective in control of SWD.

In regard to the extent of infestation (fig. 3B), the error bars (SE should be stated in Figure 3 caption, and other captions with error bars) indicated that combined years, 22MD, and 23MD are not that different as SE are overlapping considerably. For 23NJ, the push, pull, and push-pull are significantly lower (not just statistically) than control. The authors suggest that the higher infestation rates (higher populations?) in 23NJ may have allowed the MB and pull treatments to have a significant effect. Not sure why this may happen, or why the authors tried to explain this (speculation in any case).

The release rates of MB from polyethylene dispensers appear to be different (polyethylene thickness in lab vs field) between the supplementary (Fig. 1, 1000 ul is releasing 0.415 g/day and main text (line 234: 0.47 g/day). I am also wondering why the release rate was not directly proportional to the length of the polyethylene packet at least at the 1000 and 500 ul amounts (minor point of course). Because both the length and the amount were changed it is not easy to understand. Usually, amount of volatile added to the packet does not change the release rate as long as the interior is saturated. The amount of liquid volatile placed inside the dispenser instead affects the life-span of the dispenser. Several places the release in g/day or g/24 hours and in other places it is mg/hour. I think it should be in one or the other consistently in the manuscript and supplementary.

I am also wondering about the “pull” and how many SWD were caught in traps. I did not see any information about this.

More specific:

Line 150 – polyethylene pouch was 5 cm for 1000 ul, 3 cm for 500, and 2 cm for 100 ul. The release rate might be expected to be 5, 3, and 2 ratios as long as the 100 ul still saturated the inside atmosphere of pouch. However, later I see that it was, respectively, 368 mg/day, 272 mg/day, and 7 mg/h (line 331). Why not all mg/h or day (should be consistent). So 15.3, 11.3, and 7 mg/hour, respectively, which is if 15.3 = 5 then 500 might be expected to be 3/5 times 15.3 = 9.18 (rather than 11.3) and 2/5 x 15.3 = 6.12 (close to 7 mg/h). I guess this is close enough to the length of the dispenser which is according to theory. Also, these numbers are different from those shown in Supp Fig 1. Which is true?

Line 211 – can you specify what are the attractants and their proportions and dosages that were used? Please add a table that describes the lures that were used in every experiment. It is awkward to look for the attractant chemicals in the text.

Line 254-260 combine with Lines 214-217 and move figure 2 closer to those lines.

Line 272 – seems like 2 berries per plant is not so much when 3 dispensers were used on the same plant. Also, it is possible that sampler might unconsciously select an infested looking berry on a control or pull plant but a “clean” healthy berry on a push plant. I think there should have been more care in how these 2 berries were selected for sampling.

·        

·       Line 301. "sample numbers" refers to what? Number of fruits?

Line 331-335 release rates mentioned here are different from those in Fig 1 in supp.

Fig. 3 –Interesting that “pull” only worked in 23NJ, almost as well as push and together same as push. In other years pull did nothing and one could also think push did not much (Fig. 3B).

Discussion

Line 416-431 – says if at least 7 mg/hr then inhibited oviposition in lab cage (Supp. Fig. 1). But in supplemental Fig. 1 it says 1000 ul dispensers emitted 415 mg/day or 17.3 mg/h, 500 ul dispensers emitted 90 mg/day or 3.75 mg/h and 100 ul dispenser emitted 30 mg/day or 1.25 mg/h. There is a discrepancy here.

Line 457– Yes, mating disruption works less well at higher densities of moths in field, but here you have the MB working better at higher densities of SWD (the reverse). Well, so this at first seems to be contradictory, and many readers may think so. I’m not sure but you might try to explain better. At higher density of moths then males can run into a female more often and mate, so more difficult to confuse with multiple fake females and also to inhibit random interceptions. With SWD at higher population then presumable more competition for fruits and more likely to intercept at random. However, if MB is powerful the SWD may avoid, but at low densities this also should be so, so I’m not sure.

Line 471-479 –like mating disruption, mass trapping (pull) success should be higher or more effective at lower population densities, and it was not different from control in 2022 study. In 2023 in NJ (in MD there was little effect of pull) the pull (mass trapping) did significantly reduce the infestation compared to control, which should be less likely than at lower populations like in 2022. So I’m not sure about the reasoning here. Treatments reduced eggs laid in 2023 NJ, not 2023 MD or 2022 MD, but 2023 populations were higher density than 2022 yet at least in one of two sites the pull had lower numbers of eggs than control site.

Line 504 – this reminded me again that no data on pull (catches of SWD in traps). This might give an idea of the populations at different sites and plots and then it might allow better insight into speculations about densities and treatment effects of MB and attractants compared to control areas.

Line 543 – change “is” to “as”

The methods and materials are pretty well described and the presentation is good and the statistics seem well done.  

Author Response

In regard to the extent of infestation (fig. 3B), the error bars (SE should be stated in Figure 3 caption, and other captions with error bars) indicated that combined years, 22MD, and 23MD are not that different as SE are overlapping considerably. For 23NJ, the push, pull, and push-pull are significantly lower (not just statistically) than control.

(±SE) has been added to the figure caption. The statistics for the differences between treatments are presented in Table 3 and Supp Table 2. Treatments can be significantly different despite overlap in SE because the pairwise comparisons between treatments alone do not consider the effects of position and the interaction of treatment and position. To avoid confusion we decided it best to remove the lettering from the 22MD extent of infestation plot and instead simply indicate the marginally significant effect of treatment detected by the model (Table 3).

The authors suggest that the higher infestation rates (higher populations?) in 23NJ may have allowed the MB and pull treatments to have a significant effect. Not sure why this may happen, or why the authors tried to explain this (speculation in any case).

We suspect the reviewer is referring to the part of the discussion where we point out a possible correlation between pest pressure and the treatment effects we observed, so we have changed the language here to come across as less speculative. We were not trying to speculate as to why that trend might occur or claim that this trend is representative of push-pull strategies in general. We hope that the new language makes it clear that we want to point out the unexpected trend we observed and hope that further research can reveal the underlying causes. 

The release rates of MB from polyethylene dispensers appear to be different (polyethylene thickness in lab vs field) between the supplementary (Fig. 1, 1000 ul is releasing 0.415 g/day and main text (line 234: 0.47 g/day). I am also wondering why the release rate was not directly proportional to the length of the polyethylene packet at least at the 1000 and 500 ul amounts (minor point of course). Because both the length and the amount were changed it is not easy to understand. Usually, amount of volatile added to the packet does not change the release rate as long as the interior is saturated. The amount of liquid volatile placed inside the dispenser instead affects the life-span of the dispenser.

Thinner tubing was used in the field because we wanted release rates as high as possible whereas in the lab we needed release rates to be low enough to not cause mortality. The release rates listed in the caption for Supp Fig 1 were incorrect. The release rates given in the results section of the main text are correct and those given in Supp Fig 1 have been corrected. The dispensers were not saturated in the sense that they were filled to capacity, that is to say the interior of the dispensers had the liquid MB and space for air. Perhaps this was not clear because of the fact that a food saver vacuum sealer was used for their construction. We have further clarified that only the heat-sealing function and not the vacuum was used to create the dispensers. Without leaving space for air, or by using the vacuum function, there would have been an unpredictable amount of spillage while constructing the dispensers even with the presence of the felt. Only the 100 uL dispensers contained little enough liquid that the amount of felt that could fit inside the dispenser could contain the entire aliquot of MB. If the inside was filled with liquid MB and no space for air, the amount of surface area to which the liquid MB was exposed would determine the release rate and the amount of liquid MB loaded would determine the lifespan, as the reviewer stated. We scaled down both the amount of MB loaded and the size of the dispenser so that the surface area could be reduced to reduce release rates and these smaller dispensers could still be sealed without spilling the liquid MB.

Several places the release in g/day or g/24 hours and in other places it is mg/hour. I think it should be in one or the other consistently in the manuscript and supplementary.

Changed to mg/h throughout manuscript and supplementary.

I am also wondering about the “pull” and how many SWD were caught in traps. I did not see any information about this.

There was limited information provided about trap catch as the last paragraph of the results section. Further details of trap catch have been provided as Figure 4.

More specific:

Line 150 – polyethylene pouch was 5 cm for 1000 ul, 3 cm for 500, and 2 cm for 100 ul. The release rate might be expected to be 5, 3, and 2 ratios as long as the 100 ul still saturated the inside atmosphere of pouch. However, later I see that it was, respectively, 368 mg/day, 272 mg/day, and 7 mg/h (line 331). Why not all mg/h or day (should be consistent). So 15.3, 11.3, and 7 mg/hour, respectively, which is if 15.3 = 5 then 500 might be expected to be 3/5 times 15.3 = 9.18 (rather than 11.3) and 2/5 x 15.3 = 6.12 (close to 7 mg/h). I guess this is close enough to the length of the dispenser which is according to theory. Also, these numbers are different from those shown in Supp Fig 1. Which is true?

Supp Fig 1 release rates were incorrect and have been fixed. Discussion of the potential reasons for lack of proportionality as dispenser size was reduced is provided above.

Line 211 – can you specify what are the attractants and their proportions and dosages that were used? Please add a table that describes the lures that were used in every experiment. It is awkward to look for the attractant chemicals in the text.

Table added, it is now Table 1.

Line 254-260 combine with Lines 214-217 and move figure 2 closer to those lines.

 Done

Line 272 – seems like 2 berries per plant is not so much when 3 dispensers were used on the same plant. Also, it is possible that sampler might unconsciously select an infested looking berry on a control or pull plant but a “clean” healthy berry on a push plant. I think there should have been more care in how these 2 berries were selected for sampling.

Berries were only sampled if they represented berries that would be suitable both as an SWD oviposition substrate and as a berry that would be purchased by consumers. They were ripe, firm, and undamaged to the naked eye. SWD will not oviposit in green berries. Berries that lacked firmness could have already started to be consumed by SWD larvae. Noticeable damage to a berry may have affected SWD preference. By selecting only ripe, firm, and undamaged berries we avoided bias towards or against SWD damage and obtained samples representative of what U-pick customers would harvest. This was stated in the last paragraph of the field test methods section as “Unripe, damaged, or unfirm berries were not selected.” This statement has been reworded and expanded on as it is above to improve clarity.     

Line 301. "sample numbers" refers to what? Number of fruits?

 This refers to the number of independent replicates. Each fruit does not represent an independent sample, the collection of fruits sampled from each plot each week represents the independent sample. So for each treatment there are multiple reps per week and multiple sampling weeks. For example in the 22MD study, there were 5 blocks and 6 weeks of sampling, so if every treatment plot had available fruits in every block each week there would be a total N = 30. Because of losses to customers some plots and some entire blocks did not have fruits for sampling and the resulting N is less than 30 for all treatments. This wording has been clarified.

Line 331-335 release rates mentioned here are different from those in Fig 1 in supp.

Corrected

Fig. 3 –Interesting that “pull” only worked in 23NJ, almost as well as push and together same as push. In other years pull did nothing and one could also think push did not much (Fig. 3B).

Figure 3 has been changed to highlight the significant differences found in the 23NJ study.

Discussion

Line 416-431 – says if at least 7 mg/hr then inhibited oviposition in lab cage (Supp. Fig. 1). But in supplemental Fig. 1 it says 1000 ul dispensers emitted 415 mg/day or 17.3 mg/h, 500 ul dispensers emitted 90 mg/day or 3.75 mg/h and 100 ul dispenser emitted 30 mg/day or 1.25 mg/h. There is a discrepancy here.

Corrected

Line 457– Yes, mating disruption works less well at higher densities of moths in field, but here you have the MB working better at higher densities of SWD (the reverse). Well, so this at first seems to be contradictory, and many readers may think so. I’m not sure but you might try to explain better. At higher density of moths then males can run into a female more often and mate, so more difficult to confuse with multiple fake females and also to inhibit random interceptions. With SWD at higher population then presumable more competition for fruits and more likely to intercept at random. However, if MB is powerful the SWD may avoid, but at low densities this also should be so, so I’m not sure.

 Thank you for bringing up this point. The purpose of bringing mating disruption into the discussion was to try to draw a similarity between these two management strategies which have the commonality of using olfactory stimuli to disrupt behavior. If the trend in our data is truly representative of how this strategy interacts with SWD pressure, it is indeed inverse of what is observed with mating disruption. We have altered the language to make it clearer that what we observed is opposite what one might expect.  

Line 471-479 –like mating disruption, mass trapping (pull) success should be higher or more effective at lower population densities, and it was not different from control in 2022 study. In 2023 in NJ (in MD there was little effect of pull) the pull (mass trapping) did significantly reduce the infestation compared to control, which should be less likely than at lower populations like in 2022. So I’m not sure about the reasoning here. Treatments reduced eggs laid in 2023 NJ, not 2023 MD or 2022 MD, but 2023 populations were higher density than 2022 yet at least in one of two sites the pull had lower numbers of eggs than control site.

The reasoning is based on the patterns seen in the extent of infestation, where the pull treatments in 22MD and 23NJ led to greater extents compared to push or push-pull. Yes, in the 23NJ test the pull led to reductions compared to control, but that protection was not as good as the push alone and was not additive based on the push-pull result, which is explained further in the following paragraph where we discuss that it is an apparent trend with SWD that the pull agent does not provide as much protection as the push in the field. 

Line 504 – this reminded me again that no data on pull (catches of SWD in traps). This might give an idea of the populations at different sites and plots and then it might allow better insight into speculations about densities and treatment effects of MB and attractants compared to control areas.

We have included a new figure that shows weekly trap catch and infestation rates (Fig 4) and expanded the discussion of trap catch results.

Line 543 – change “is” to “as”

Done

Reviewer 2 Report

Comments and Suggestions for Authors

The manuscript “Evaluation of a Push-pull Strategy for Spotted-wing Drosophila Management in Highbush Blueberry Orchards” by Cody Gale, Beth Ferguson, Cesar Rodriguez-Saona, Vonnie D. Shields and Aijun Zhang presents an interesting study on the field effects of methyl-benzoate on oviposition behaviour of D. suzukii.  Control of D. suzukii is still an open question as the current methods are for different reasons insufficient. Therefore, the paper presents an interesting contribution to the development of future control mechanisms.

However, from my point of view, a combination of experiments with largely different results into one statistical model, which then points out a significant treatment effect, is no expedient approach. It is a pity, that a second experiment with high infestation pressure is lacking, but it is as it is, and this should be discussed. In general, I believe that the presentation of more detailed data together with the already presented outcome of the statistical models (e.g. data for individual sampling dates in relation to e.g. the climatic conditions for the experiment 23NJ) would provide a better insight into treatment effects.

I had difficulties to follow the exact plans and schedule of the field experiments. Maybe this could be simplified by the presentation of a table of all experiments, summarizing the description in page 6 (and, also presenting sampling dates) or by dividing the description of the field tests into sub-chapters.

Some comments in detail are outlined below:

L 21-23 -  from a practical point of view the effect of the treatment is still not sufficient. It might be promising and become a component of an overall strategy but infestation rates of more than 50% are not acceptable for the growers.

L 32 from my point of view it is not correct to pool 2 experiments without effect with one with an effect.

L36-37 see above

L 134 does set of no choice assays mean that the assays were carried out at different dates with different batches of blueberries? How many experiments with how many berries per set?

L 271 samples were collected how many times?

L 287 in each case with a corresponding control treatment?

L 291 date(s) not outlined in MM

L 305 or?

L 331-343- data on oviposition to be illustrated first?

L 344-353 and Figure 3. In each year, the field trials lasted several weeks and data were collected weekly. However, the results only present outcomes of an overall statistical model including all sampling dates and all blocks. At the same time (for 2022, for 2023 no data available?), heavily fluctuating climatic conditions are reported. It must be presumed that the fluctuations in temperature and wind speed also affected the fly populations and thus the outcome of the current experiment? At least for the experiment 23NJ split data should be reported? Was there an impact of higher wind velocity on the effect of the dispensers, did high temperatures reduce the fly populations? Did the block numbers significantly affect the outcome?

L 393-406 no expedient compilation

L 408 -411 Data represent an average of all field experiments ? Data should be reported individually for each trial ?  General comment: Low overall fly populations or low attractiveness of the lure traps??

L 412 -414 not aims of the current study?

L 415 and ff avoid repetition of results

L 480 and ff. pulling component not very effective – trap catches are low?

Author Response

However, from my point of view, a combination of experiments with largely different results into one statistical model, which then points out a significant treatment effect, is no expedient approach. It is a pity, that a second experiment with high infestation pressure is lacking, but it is as it is, and this should be discussed. In general, I believe that the presentation of more detailed data together with the already presented outcome of the statistical models (e.g. data for individual sampling dates in relation to e.g. the climatic conditions for the experiment 23NJ) would provide a better insight into treatment effects.

The analysis of the combined data has been removed. New language has been added to the discussion to point out the need for further testing under high pest pressure, such as, “Since only the one field test produced significant results for treatment differences, further testing in high pest pressure scenarios is needed to better understand this trend.” The comment on climatic conditions is addressed below.

I had difficulties to follow the exact plans and schedule of the field experiments. Maybe this could be simplified by the presentation of a table of all experiments, summarizing the description in page 6 (and, also presenting sampling dates) or by dividing the description of the field tests into sub-chapters.

Table 2 now provides details of the activities performed each week in the field tests.

Some comments in detail are outlined below:

L 21-23 -  from a practical point of view the effect of the treatment is still not sufficient. It might be promising and become a component of an overall strategy but infestation rates of more than 50% are not acceptable for the growers.

Ok, this point has been added to the simple summary.

L 32 from my point of view it is not correct to pool 2 experiments without effect with one with an effect.

Combined data removed

L36-37 see above

Combined data removed

L 134 does set of no choice assays mean that the assays were carried out at different dates with different batches of blueberries? How many experiments with how many berries per set?

Correct, these sets of experiments took multiple weeks to complete. A fresh pack of blueberries was purchased at the beginning of each week. Typically 2-4 experiments (4-8 independent replicates for control and treatment) were performed in a week. This information has been added to the text. Experiments were always performed with controls and treatments simultaneously. This was stated in the methods but has been reiterated there for clarity.

L 271 samples were collected how many times?

Table 2 now shows each week that samples were collected for each field test.

L 287 in each case with a corresponding control treatment?

Yes, this has been clarified in the text.

L 291 date(s) not outlined in MM

Data for controls and treatments were always generated simultaneously. It took multiple weeks to carry out all of these tests, but a fresh pack of blueberries was used each week, and the flies from the colony were always of a specified age range. This has been clarified in the text. The exact dates on which these experiments were conducted is not pertinent information given the highly controlled laboratory settings.

L 305 or?

Frequency and extent of infestation were analyzed separately. This has been reworded for clarity.

L 331-343- data on oviposition to be illustrated first?

Emission rates of the dispensers were determined as part of the experiments and are thus presented here in the results. These rates are not illustrated in a figure as we feel the linear equations are sufficient. The data on oviposition in these lab tests is illustrated only in a supplemental figure because the primary focus of the manuscript is on the field tests.

L 344-353 and Figure 3. In each year, the field trials lasted several weeks and data were collected weekly. However, the results only present outcomes of an overall statistical model including all sampling dates and all blocks. At the same time (for 2022, for 2023 no data available?), heavily fluctuating climatic conditions are reported. It must be presumed that the fluctuations in temperature and wind speed also affected the fly populations and thus the outcome of the current experiment? At least for the experiment 23NJ split data should be reported? Was there an impact of higher wind velocity on the effect of the dispensers, did high temperatures reduce the fly populations? Did the block numbers significantly affect the outcome?

We do not have weather data for the field tests. The weather data reported in 2022 was generated specifically for determining the release rate of MB dispensers in the field. That data was generated at the Beltsville Ag Research Center where we have access to a weather station and the data were generated just prior to the start of the 2022 field season. The purpose of including the weather data relevant to the release rate tests was to demonstrate that the release rate was determined under highly variable field conditions and thus is an accurate estimation for what to expect during the field tests. Weekly data on overall trap catch and infestation are now presented in Figure 4. There was no difference in pull versus push-pull trap catch, and thus no reason to separate these data by treatment. The most appropriate statistical approach is to assess the effects of treatment, position, and their interaction on the two measures of infestation that we collected using sampling date and block as random factors influencing the response. With a frequentist statistical approach there is not a clear valid reason to assess the significance of random effects as they are not truly parameters of the model of interest. The experimental design incorporates blocks to achieve the necessary replication and the model incorporates the underlying variation due to block as a random effect influencing the parameters of interest. Comparing a model that includes block and a random effect to a model that doesn’t will conclude that the models are significantly different, but this is to be expected, and if anything only speaks to the validity of including block as a random variable in the final statistical model used.

L 393-406 no expedient compilation

Combined data removed

L 408 -411 Data represent an average of all field experiments ? Data should be reported individually for each trial ?  General comment: Low overall fly populations or low attractiveness of the lure traps??

More details on trap catch have been reported in Figure 4.

L 412 -414 not aims of the current study?

Unclear what the reviewer is asking. If asking if the aim of the current study was to assess the trap catch obtained by the quinary blend used, no. See Larson et al 2021, as the purpose of that study was to assess the blend in multiple fruit crops in various geographic locations.

L 415 and ff avoid repetition of results

Repeated results from the lab tests were removed from the first paragraph of the discussion. Results from the 22MD test are reiterated in what is now the third paragraph of the discussion for the purpose of providing the readers with a clear narrative for why the 22MD test differed from the 2023 tests. Results are reiterated in what is now the fourth paragraph of the discussion for the purpose of providing clarity to the readers when we point out the trend for our treatments having the strongest effect at the highest pest pressure. We have further clarified that we do not suppose to understand why this is and that further testing is needed to understand this trend.

L 480 and ff. pulling component not very effective – trap catches are low?

The pull component was not effective in the sense that it did not provide as much crop protection as the push component alone and did not provide additive protection when used in conjunction with the push. This has been clarified in the text.